# Electronic Structure and Solvation Effects from Core and Valence Photoelectron Spectroscopy of Serum Albumin

**DOI:** 10.3390/ijms23158227

**Published:** 2022-07-26

**Authors:** Jean-Philippe Renault, Lucie Huart, Aleksandar R. Milosavljević, John D. Bozek, Jerôme Palaudoux, Jean-Michel Guigner, Laurent Marichal, Jocelyne Leroy, Frank Wien, Marie-Anne Hervé Du Penhoat, Christophe Nicolas

**Affiliations:** 1Université Paris-Saclay, CEA, CNRS, NIMBE, CEA Saclay, 91191 Gif-sur-Yvette, France; lucie.huart@cea.fr (L.H.); laurent.marichal@grenoble-inp.fr (L.M.); jocelyne.leroy@cea.fr (J.L.); 2Synchrotron SOLEIL, 91192 Saint Aubin, France; aleksandar.milosavljevic@synchrotron-soleil.fr (A.R.M.); john.bozek@synchrotron-soleil.fr (J.D.B.); frank.wien@synchrotron-soleil.fr (F.W.); 3Institut de Minéralogie, de Physique des Matériaux et de Cosmochimie, Sorbonne Université, UMR CNRS 7590, MNHN, 75252 Paris, France; jean-michel.guigner@sorbonne-universite.fr (J.-M.G.); marie-anne.herve_du_penhoat@sorbonne-universite.fr (M.-A.H.D.P.); 4Laboratoire de Chimie Physique-Matière et Rayonnement, Sorbonne Université, UMR CNRS 7614, 75252 Paris, France; jerome.palaudoux@sorbonne-universite.fr

**Keywords:** X-ray electron spectroscopy, protein, electronic structure, theoretical chemistry, hydration

## Abstract

X-ray photoelectron spectroscopy of bovine serum albumin (BSA) in a liquid jet is used to investigate the electronic structure of a solvated protein, yielding insight into charge transfer mechanisms in biological systems in their natural environment. No structural damage was observed in BSA following X-ray photoelectron spectroscopy in a liquid jet sample environment. Carbon and nitrogen atoms in different chemical environments were resolved in the X-ray photoelectron spectra of both solid and solvated BSA. The calculations of charge distributions demonstrate the difficulty of assigning chemical contributions in complex systems in an aqueous environment. The high-resolution X-ray core electron spectra recorded are unchanged upon solvation. A comparison of the valence bands of BSA in both phases is also presented. These bands display a higher sensitivity to solvation effects. The ionization energy of the solvated BSA is determined at 5.7 ± 0.3 eV. Experimental results are compared with theoretical calculations to distinguish the contributions of various molecular components to the electronic structure. This comparison points towards the role of water in hole delocalization in proteins.

## 1. Introduction

The electronic structure of proteins controls some of their major biological processes, such as protein–ligand interactions [1], cofactor properties [2], and, more specifically, charge transfer in biosystems. Indeed, several electron transfer mechanisms, i.e., electron hopping and super-exchange, have been proposed since the pioneering studies of Vault and Chance and Gray and Winckler [3,4]. These models involve electron tunneling, where an energy barrier is defined either in the valence band (hole transfer) or conduction band (electron transfer). These barrier values are critical for calculating tunneling efficiencies and have up, until now, been based on crude assumptions, gas-phase spectroscopic measurements, or electronic calculations of simple model compounds [5]. Published values range from −10 eV to −4 eV [6].

The use of a liquid micro-jet to study solvated molecules with electron spectroscopy emerged in the 1980s, with the pioneering work of Faubel and co-workers [7]. Although electron spectroscopy was first performed on liquids [8] in the 1970s, only low vapor pressure liquids were amenable to study. In-vacuum liquid micro-jet techniques allow studying liquids with high vapor pressure (>4 mbar), such as water and aqueous solutions. A wide variety of solutions has been studied, from salts [9] to small biological molecules [10,11,12,13], and even nanoparticles [14,15] using synchrotron radiation. In most of these studies, the solution is collected on a cold trap after ionization and lost for further analysis. If the cold trap is replaced by a heated and pumped catcher, the “spent” liquid jet is collected outside the vacuum chamber of the spectrometer. In that case, the sample solution can be subjected to additional analytical analysis. Here we propose a new application of liquid micro-jet electron spectroscopy to studies of large biomolecules, such as proteins. Bovine serum albumin (BSA) is well suited for this proof of concept experiment due to its affordability and well-known properties [16]. Moreover, BSA has been studied previously by X-ray photoelectron spectroscopy (XPS) deposited on surfaces and, more recently, dissolved in a salty water droplet using a near-ambient pressure spectrometer [10]. New insights into the electronic structure of BSA and the effects of solvation were obtained here by the photoelectron spectroscopy of the solution in a liquid micro-jet combined with ab initio electronic state calculations.

## 2. Results and Discussion

### 2.1. Stability of the Protein in the Liquid Jet and under Irradiation

Vacuum/gas/liquid interfaces, present in this experiment, have been shown to alter the proteins structure [17,18], although, our experimental conditions are expected to limit this phenomenon [19]. Therefore, the sample was analyzed before and after its passage through the liquid jet XPS experiment (described in Section 3 and Appendix A) to examine it for possible structural changes. The original and recovered solutions were examined for protein fragmentation using SDS PAGE gel electrophoresis (see Figure 1A). The samples were deposited with increasing protein concentration per well (dilution ratio 1:8, 1:4, 1:2, and 1). In addition to the expected monomeric form of the BSA (~66 kDa), we observe multimeric forms (>130 kDa) and some fragments (<15 kDa) for the low dilution samples. Their presence in both the original and collected samples suggests that they are not the result of damage or aggregations from the liquid micro-jet experiment but rather from the preparation of the samples.

UV–SRCD spectra of the protein solutions before (blue) and after (red) passing through the liquid jet setup were compared to check for the preservation of the protein’s secondary structure (Figure 1B). SRCD allows measurements at high concentrations, such as those used here. At this concentration (~36 g/L), the protein is self-buffered (pH = 7.2). We thus expect a stable folding and homogeneous distribution of the protein in the solution (19). However, should any damage have occurred due to the liquid jet environment or synchrotron radiation exposure, a change in the folding is expected. The comparison of SRCD spectra obtained from the injected sample and the collected one does not show significant differences, as shown in Figure 1B. The high content of helix structures (~70%) is in good agreement with the reported structure of the BSA [20,21] (see structural values in Appendix A). Therefore, no modification of the protein was detected, which confirms the thesis that the protein remains folded in its well-known heart shape [22].

### 2.2. Core Ionization Study

Contrary to the valence band, XPS spectra of BSA have been previously reported, mainly in a solid state (Appendix A), although one study was performed on BSA [10] in solution. These previous works disagree on the number of bands assigned to the XPS spectra. To better understand these differences, BSA core-level spectra recorded in both solvated and solid states are presented in Figure 2A for the carbon (C1s) and in Figure 2B for the nitrogen (N1s) levels. Results for the oxygen 1s level are given in the Appendix A, although the protein contribution is hidden by contributions from liquid water.

Throughout the rest of the article, XPS solid-state spectra are presented in binding energies (given with respect to the gold Fermi level), whereas XPS liquid micro-jet spectra are presented in ionization energies (referenced to the vacuum level at infinity rather than the Fermi level, as explained in Section 3 and Appendix A*)*.

The N1s spectrum of the solution (Figure 2B) exhibits one major peak at 406.3 eV, with a shoulder at 404.9 eV. The peak around 406.3 eV is attributed to nitrogen from peptide bonds and that at 404.9 eV to unprotonated amine side chains. These two peaks are preserved in the solid-state spectra but with a shift in energy.

The energy difference between the peaks positions in the solid and solution result from the work function of the solution [23]. The work function of the BSA solution thus obtained is Φ_solution_= 4.9 (±0.2) eV. This value is similar to that reported in the literature for the water of around 4.7 eV [24,25] and seems to validate the observation that the work function of aqueous solutions is relatively independent of the solute [23].

In the C1s spectrum, three peaks were identified in both the solid and solution spectra. The values resulting from the fit of these three peaks (positions and widths) are presented in Table 1 and compared to those obtained by Jain et al. using NAP-XPS [10]. The choice to use only three contributions is based on the higher constraints imposed by the high resolution XPS spectra obtained. Indeed, a fit with a fourth contribution will result, in our case, in areas that are not representative of the proportions of the most abundant types of carbon (see Table 1). Moreover, this choice is also supported by our calculations (see discussion below). The peak around 289.8 eV is attributed to hydrocarbon carbons (only bonded to carbon and hydrogen); that at 291.1 eV to carbon singly bonded to oxygen or nitrogen; while carbonyl and/or amide carbon and carboxyl carbon are regrouped in the peak centred around 293.0 eV. These ionization energies are consistent with values obtained on hydrocarbons by Björneholm et al. [26,27]. The same work function of the BSA solution is obtained at the C1s edge as at the N1s edge.

The high-resolution C1s photoelectron spectra reported here, with three distinct peaks, are different from the results of Jain and co-workers where the C1s peak envelope was fit with four symmetric peaks corresponding to carbon bonded to carbon and hydrogen (285 eV), carbon singly bonded to oxygen or nitrogen (286.2 eV), carbonyl and/or amide carbon (287.4 eV), and carboxyl carbon (288.6 eV) [10]. These measurements highlight the fact that deconvoluting XPS signals in such large molecules is complicated by the difficulty in deciphering contributions from atoms in different chemical environments. The origin of differences in binding energies as a function of the chemical environment is considered to explain the discrepancy. Considering the most abundant types of carbon in a protein, three major bands are expected: C_alpha_ (583 of 3181 carbon atoms in BSA), C_beta_ (567/3181), and C_peptidic_ (583/3181). The C_aliphatic_ (coming from the valine, leucine, and isoleucine side chains) are the fourth more abundant population (302/3181), followed by additional minor bands for C_carboxylate_ (99/3181), C_alcohol_ (62/3181), etc. The binding energy, E_prot_, of each of the different species of atoms is related to the binding energy $E_{\mathrm{vac}}^{\infty }$ of an isolated atom of the same electronic state in vacuum by:$$E_{\mathrm{prot}}=E_{\mathrm{vac}}^{\infty }+\varepsilon \;Q+V+E_{\mathrm{pol}}$$
where Q is the charge of the atom considered, ε is core-frontier coupling constant, V is the potential energy of the electron to be photoionized in the electrostatic field generated by the distribution of charges in the molecule (sometimes designated as the Madelung constant), and E_pol_ is the polarization energy of the system, when the core hole is formed after photoemission. Therefore, Q and V are the initial state effects, whereas E_pol_ is a final state effect. Q depends on the chemical environment of the atom. It is the main determinant of the chemical shift in organic molecules and can be obtained from ab initio calculations on the initial state. V depends on the location of the atom within the protein, as the electrostatic potential is not homogeneous. It is assumed that all atom types will feel the same potential V within each amino acid.

The dependence of E_pol_ on the atom types/atom location has been explored by ab initio calculations of the core ionized final state. Relaxation effects cannot be explored systematically by calculation in a macromolecule since there are as many final states as carbon atoms in the protein. The intramolecular relaxation energy term is, however, nearly constant for most atoms in organic compounds, and therefore initial state effects dominate C1s binding energy shifts [28,29]. The shape and position of the XPS signals are therefore analyzed using the information on the Q and V distributions only. The Density Functional Theory (DFT) was used to calculate the atomic charges Q of the BSA structure with a 3 Å water layer on its surface (BSA_wat_), as presented in the Appendix A (computational details are described in Section 3). In such a system, the number of atoms coming from the protein and from the added layer are equivalent. The Hirshfeld description of charges was used, which seems to be more relevant in the XPS context [30,31]. The distribution of carbon charges is spread out over four major populations: the peptidic carbons (0.4 ± 0.1 e), the alpha carbons (−0.2 ± 0.4 e), the aromatic (−0.4 ± 0.2 e), and the aliphatic carbons (−1.2 ± 0.5 e), as illustrated in Figure 3A. Hydration leads to very limited changes in the charges. Indeed, the comparison of Hirschfeld charge calculations with water versus without added water displays a slope near one that substantiates our affirmation (see the Appendix A). It is clear that this level of description is incomplete and that both the number of bands and their widths are not adequately represented. However, it is also clear that a significant component cannot be attributed solely to carboxylate. Therefore, the effect of V was considered in addition to that of Q using standard electrostatic calculations for proteins in solution [32].

All Q charges were used for this calculation. The electrostatic field (expressed in k_b_T/e) was then calculated at various atomic positions, allowing both local (Q) and global (V) contributions to be taken into account. This combination captures the main features of the carbon spectra with only three main bands, as presented in Figure 3B. The first two can be attributed, respectively, to C_peptidic_ (at 293 eV) and C_alpha_ (at 291.1 eV) but the third one (at 289.8 eV) comes from both C_beta_ and C_aliphatic_ carbons. Contributions from the aromatic amino acid carbons are distributed beneath the two remaining bands. As stated earlier, these results highlight that the XPS spectra interpretation of such a macromolecular system requires guidance from theoretical support to eventually disentangle the different contributions.

### 2.3. Valence Band Edge Analysis

The value of the valence band threshold for a protein in solution is reported here for the first time (Figure 4 and Appendix A). The top of the valence band is clearly dominated by the signal of the water 1b1 molecular orbital (band III on Figure 4). It could be fitted by a Gaussian function centered at 11.4 ± 0.1 eV with a full width at half maximum of 1.76 eV (Appendix A), yielding a photoionization threshold for liquid water of 10.0 eV by the extrapolation of the 1b1 band slope (see linear fit on Appendix A). This value is within the measurement error of the value reported in the literature, 9.9 eV [25,33]. Therefore, the solvated protein has a minimal effect on the structure and work function of the water solvent. Additional bands at lower binding energy can be attributed to the protein. Indeed, two steps are observed in the spectrum below the 1b_1_ band of water (the bands I and II of Figure 4). The corresponding thresholds determined by extrapolating the leading slope of the steps, as shown on the inset of Figure 4, were found to be at 5.7 ± 0.3 eV and 7.4 ± 0.1 eV.

Precise band gap calculations of proteins are still rare, but the absolute value of the calculated valence band onset for peptides (−7 eV [34,35] below vacuum level) is significantly lower than the one measured here (−5.7 eV), considering that the energy of the highest occupied molecular orbital (HOMO) value is the opposite of the ionization threshold. Even the readily oxidizable amino acid monomers (tryptophan and tyrosine) have HOMO energies in the -8 eV range [5,35]. In addition to aromatic amino acids, we can envision three origins for this upper part of the BSA valence band. It can come from the thiol functions. Indeed, HOMO energies of various types of cysteinates have been measured in the −5.5 eV range [36,37]. However, there is only one free cysteine in BSA [38] and the measurements are not expected to be so sensitive. Secondary structures, i.e., alpha helixes, the main structural motif in BSA, can significantly alter the valence band energy (up to 2.5 eV increase of the HOMO energy [35,39,40]). Finally, electrostatic effects can also be envisioned [41].

In order to decipher the atomic contribution of the upper part of the valence band, electronic structure calculations were performed on a BSA structure with a 3 Å water added layer, as presented in the Appendix A. We are aware that precise calculations on such large systems are still in the early stages of development [40], but modern codes can give some interesting guidance in interpreting the experimental spectra [41]. The projected density of states (PDOS) for carbon, nitrogen and sulfur atoms are represented in Figure 5A. The data for oxygen and hydrogen atoms are not represented here, as the water contribution overwhelms that of the BSA.

The sulfur PDOS is barely visible in Figure 5A, with a small maximum around −4.2 eV. This indicates that the cysteine of the proteins, even if they can be part of, are not the main component of the protein HOMOs. Carbon, which is the highest contributor to the density of states, shows two major peaks at −4.7 eV and −7.6 eV binding energy. These seem to contribute, respectively, to bands I and II of the experimental spectra (represented in red). Band I of the valence spectra also has contributions from the nitrogen PDOS (around −5 eV). This suggests that the orbitals in this energy region have a significant contribution from carbon/nitrogen functional groups, either peptide bonds or aromatic amino acids, such as histidine and tryptophan. Band II therefore arises mainly from CHx groups. In order to confirm the underlying contribution of the carbon PDOS, three additional projections were conducted, on the methyl carbon of alanines (noted Cbeta in the Appendix A), the peptidic carbon of alanines (noted Cpeptidic in the Appendix A), and the delta carbon of aromatic amino acids (named Carom in the Appendix A). Alanines were chosen because of their 46 occurrences in BSA distributed throughout the protein. Irrespective of the secondary structures the Cpeptidic belongs to (at the end or in the middle of alpha helices, in turn) their PDOS have a contribution in the −5 eV range. The alpha helix structure motif does not impose a dramatic shift in this DOS but rather a widening that suggests some delocalization of the corresponding orbitals. This contribution remains clearly distinct from that of the Cbeta PDOS that lies in the −7.6 eV range. As the Carom contributions lie also in the −5 eV range (the Appendix A), we will call band I the “unsaturated” band and band II the “saturated” band in the rest of the paper.

Although the energy distribution of the PDOS seems to agree with the experimental result, the intensities of the respective bands differ. Considering only the carbon contribution, the ratio between the saturated and unsaturated bands is around 3, while in the experimental valence spectra, the ratio is ~50. It must be noted that the oxygen PDOS was not calculated here, because of the cumbersome theoretical calculations. However, oxygen atoms are present in almost the same amount as nitrogen atoms (~19%At [42]. Their PDOS, therefore, should not be the only explanation for the ratio difference observed. Photoelectron data may differ from the total density of states for four reasons: (1) signal broadening, by a decrease in the photoelectron scattering lifetime (2) difference in ionization cross-section, (3) photoelectron inelastic scattering, and (4) inhomogeneous broadening connected to the formation of polarons by the relaxation of the atomic position around the formed holes. This is the major source of broadening in molecular systems [43,44]. The first case applies mainly to metals [45]. For the second case, in the Gelius interpretation [29,46,47], differences in molecular cross-sections result from the atomic cross-sections. C, O, and N cross-sections, however, are not so different at the considered energy. Furthermore, differences of the cross-section of the atomic subshells (s, p) are neither an explanation, as in the considered photon energy region (100 eV), their photoionization cross-sections are very close [48,49]. For the third case, differences in photoelectron scattering between saturated and unsaturated bands may at first seem to be a more appropriate explanation. One can imagine that peptide bonds, especially in alpha helices, may be deeper in the protein than the amino acid side chains surrounding them. Thus, their photoelectrons are more scattered. However, the calculated distances from the surface of various types of carbons (Appendix A) do not exhibit any significant differences.

For the fourth case, solvation is a significant source of inhomogeneous broadening, as it provides a fluctuating electronic environment through H-bonds network. Furthermore, water is known to play a role in polaron stabilization [50].

A valence band spectrum of dry BSA was measured to investigate this effect. Without the overwhelming contribution of water, six features/bands were observed (see Figure 5B). Band I was fitted by a Gaussian function centered at 4.3 ± 0.35 eV and the ionization threshold was extrapolated to a value of 2.5 ± 0.35 eV. However, it is difficult to evaluate the influence of the substrate on the deposited dried BSA.

The ratio of the intensity of bands I and II from the experimental valence spectra is now around 1.4. This value is closer to the one expected from the DOS calculations, of around 1.7. The dehydration of the protein seems to only make the unsaturated band more apparent. Therefore, to explain the low intensity of the unsaturated band in the hydrated protein, we can propose a change in the bandwidth due to a polaron formation favored by the presence of water. To increase the width of the band sufficiently to account for the difference in intensity between the solvated and unsolvated states, the band I has to be broadened by a factor of twenty in water. This would correspond to a linewidth of 2–3 eV in the hydrated case, considering the usual linewidth of 0.1–0.15 eV observed in gas phase XPS [51]. Such a linewidth is compatible with the calculation made for putative polaronic states in proteins [52,53]. Water can impact these polaronic processes through its intrinsic dielectric properties but also by activating specific vibrational modes in the protein structure [54].

### 2.4. Implications of Ionization Energies and Work Functions of the Hydrated BSA

The ionization energy threshold (IET) of the band I of the BSA with respect to the vacuum level in the liquid jet measurement is IET_vac, Band I(Liquid)_ = 5.7 ± 0.3 eV. Combined with the evaluated BSA solution work function (Φ_solution_~4.9 ± 0.2 eV), this yields a remaining energy difference between the Fermi level and the HOMO of BSA of 0.8 eV. The UV–vis spectrum of BSA (Appendix A) completes this picture with an optical band gap value of 4.25 ± 0.07 eV for the liquid state (see Figure 6). The comparison between BSA energy diagrams in both liquid and dry states is in favor of a significant electron depletion in the case of the solvated protein, as its Fermi level is lower than the expected 2 eV for a bandgap of 4 eV.

These results, therefore, shed new light on the role of water in protein conduction. Water is known to favor protein conductivity [55]. This effect was described as the opening of a protonic conduction pathway [56,57] by water. Even if early articles identified in parallel an electronic pathway activated by water [55], the current results suggest that the electronic components of protein conductivity occur through hole transport, as previously suggested by pulse radiolysis studies [58]. Along with the change in the valence band shape observed upon hydration, these observations point towards a specific role of water in the hole delocalization in proteins.

## 3. Materials and Methods

### 3.1. Chemicals

Lyophilized Bovine Serum Albumin (Sigma A0281, Merck KGaA, Darmstadt, Germany) was dissolved at 36 g/L and dialyzed in pure water. The solution was centrifuged at 14,000× *g* for 5 min at 4 °C and filtrated (0.22 µm) before use. The pH of 7.2 was measured at 20 °C using a Mettler Toledo S 220 pH meter with an InLab Nano electrode. The final protein concentration was determined by UV–vis spectroscopy (Shimadzu UV-2450 spectrophotometer, Shimadzu Corporation, Kyoto, Japan) using a molar extinction coefficient of 43,824 M^−1^ cm^−1^ at 280 nm [59]. After dialysis and centrifugation, it was determined to be ~36 g/L, which increased to 49 g/L after passing one time through the in-vacuum liquid jet due to water evaporation.

### 3.2. Photoelectron Spectroscopy

Photoelectron spectroscopy measurements were performed using two different apparatus. Liquid jet experiments were carried out at the PLEIADES beamline of SOLEIL Synchrotron, Saint-Aubin, France. The liquid jet setup is described in detail in a previous publication [60] and is shown schematically in Appendix A. Photoelectron spectra were collected using a vertically mounted VG-Scienta model R4000 WAL electron energy analyzer. The spectrometer, liquid jet, and synchrotron light axes were mutually orthogonal, as shown in Appendix A. The polarization vector of the linear polarized light was set to the magic angle (54.7°) relative to the spectrometer axis. The total experimental resolutions of 0.03 eV, 0.13 eV, and 0.2 eV were used at the selected photon energies of 100 eV, 400 eV, and 600 eV, respectively. Spectra recorded at 100 eV and 400 eV were calibrated against the adiabatic ionization energy of the X 1B_1_ state of gas phase [61] H2O^+^ at 12.621 eV and for those recorded at 600 eV, the O1s level at 539.89 eV [62,63,64]. Gas–phase spectra were obtained in the same experiment using signal obtained far from the liquid jet to avoid possibly disturbing electric fields (see Appendix A). While in operation, the pressure in the liquid micro-jet chamber was ~10^−4^ mbar, while the differentially pumped spectrometer chamber was ~8 × 10^−6^ mbar. The protein solution was flowing through a glass-nozzle (Φ~60 µm, flow rate~2.2 mL·min^−1^ or ~12 m·s^−1^, T~283 K). A metallic connector, inserted 20 cm upstream of the glass-nozzle in the PEEK injection line, was used to ground the liquid jet electrically. After crossing the synchrotron light, the liquid jet was collected in a copper-beryllium catcher with a 300 µm diameter inlet. The base of the catcher was heated to ~80 °C to avoid freezing of the liquid. The liquid temperature at the exit of the catcher remained below 35 °C. The sample solution was collected externally in a pumped glass bottle (<9 mbar) and stored at 4 °C for further analysis.

Solid–phase electron spectra were measured with a Kratos AXIS ultrahigh resolution spectrometer with a monochromatic Al K_α_ X-ray source [65]. Samples were diluted in water and dried on a gold-coated glass plate. Protein flakes were directly deposited on copper tape to avoid gold contribution for valence photoelectron spectra. The energy scale was calibrated using the Au4f_7/2_ at 84.0 eV for gold substrate and on adventitious carbon at 284.9 eV for measurements on copper. Resolution for valence spectra recorded with a pass energy of 20 eV was 0.35 eV and 0.5 eV for XPS spectra recorded with a pass energy of 40 eV. Two spectra of a C 1s inner-shell were performed on solid samples to check the stability of the sample and verify the absence of degradation during acquisition.

XPS solid-state spectra are presented in binding energies (given with respect to the gold Fermi level) as the sample is deposited on a gold substrate in electrical contact with the analyzer. The alignment of Fermi levels is verified by the consistent values of the hydrocarbon C1s line at 284.9 ± 0.5 eV [24] for the two independent studies. Energies in the liquid micro-jet spectra are referenced to the vacuum level at infinity rather than the Fermi level, even though the conductivity of the solution ensures the electrical contact with the analyzer (liquid grounded) (see Appendix A). The energy difference between the peaks in the solid and solution results from the work function of the solution [23].

XPS-spectra were fit using the commercial software package CasaXPS [66,67]. All peak fittings were performed using a sum contribution Voigt-type function with 30% Lorentzian (SGL [30]). Backgrounds were subtracted using a Toutgaard function [66,67].

### 3.3. Structural Characterization

The protein solution collected after one passage through the liquid microjet (irradiated at 400 eV (~3 × 10^12^ ph/s)) was analyzed and compared to the original solution (see Appendix A). Fractions were separated by sodium dodecyl sulfate-polyacrylamide gel electrophoresis (SDS PAGE) (12%). UV synchrotron radiation circular dichroism (SRCD) spectra in the spectral band from 170–260 nm were collected at the DISCO beamline at Synchrotron SOLEIL [68,69]. The path length was measured by the light interference of an empty cell on a spectrophotometer in the visible domain. Spectra were averaged, baseline subtracted, and normalized with a standard ((+)camphor sulfonic acid) using CDtoolX software [70]. Secondary structure contents were determined using the online server BeStSel [69,71].

### 3.4. Computational Details

The initial geometry of the protein was taken from the protein data bank (4F5S [72]). The topology of the BSA monomer was built using the GROMACS software [73]. In one configuration, acidic and essential amino acids were charged following the suggestion of the software in order to achieve a neutral global charge. The protein was then encased with a 3 Å water layer to represent the solvated state. The addition of a 3 Å water layer ensured conditions for the formation of a first continuous hydration shell [74]. The layer was added using GROMACS, ensuring the precision in additions. We considered that this layer captured the main feature of hydrogen bond and water reorientation effects [75] and that the rest of the solution could be accounted for using solvent continuum approximation. In a second configuration, acidic and basic residues were neutralized without any solvation to represent the solid-state. These structures were then energy minimized and used for further calculations using the CP2K software [41].

The Perdew–Burke–Ernzerhof (PBE) density functional was used in all calculations with a Gaussian and plane waves dual basis set and GTH pseudopotentials [76,77]: the Gaussian part was the MOLOPT-DZVP-SR-GTH basis set (double-ζ Gaussian, with one polarization function optimized for molecular systems) and the plane wave was expanded up to a density cut-off of 400 Ry.

## 4. Conclusions

This study demonstrated that liquid micro-jet photoelectron spectroscopy is a powerful tool to retrieve valence and inner-shell XPS spectra of large biomolecules, such as proteins, without damaging the sample. Moreover, it also establishes a methodology to determine, for a protein, the effect of the solvation on its electronic structure. This is of prime importance for biological processes, such as protein–ligand interactions [1], cofactor properties [2], and, more specifically, charge transfer in biosystems. Dry BSA samples exhibited XPS spectra with similar shapes. The decomposition of the spectra into various components cannot be carried out as for small molecules. Indeed, the large structure of a protein imposes the inhomogeneous broadening of the XPS signal. The heterogeneous charge repartition inherent to its structure (hydrophobicity functions and amino acids’ electrostatic charges) will affect the XPS signal position.

From the photoemission experiment, a vertical ionization energy of 5.7 ± 0.3 eV was obtained for the solvated BSA. The intensity ratio of valence bands, coupled with theoretical calculations and band gap measurements, point towards the specific role of the water in hole delocalization in proteins. This is expected to be a general phenomenon for other proteins. Valence band analysis paves the way for more complex experiments on protein electronic structure, such as by resonant Auger and core hole clock spectroscopies, for example, in order to probe the lowest unoccupied molecular orbitals and to study the different dynamics of de-exitations of the exited core electron.

## Figures and Tables

**Figure 1 ijms-23-08227-f001:**
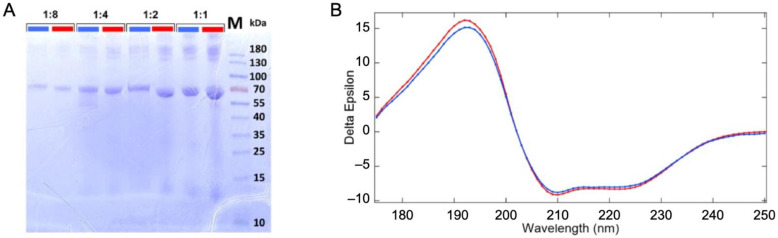
Sample analysis before and after passing through the liquid jet system and the irradiation by soft X-ray photons. (**A**) SDS PAGE analysis of the BSA solution fractions. Comparison of the original solution before injection through the liquid jet (blue) and the collected solution after irradiation with synchrotron beam (red) with different dilution factors of 1:8, 1:4, 1:2, and 1:1. (**B**) Synchrotron radiation circular dichroism spectrum of the injected (blue) and the collected solutions (red).

**Figure 2 ijms-23-08227-f002:**
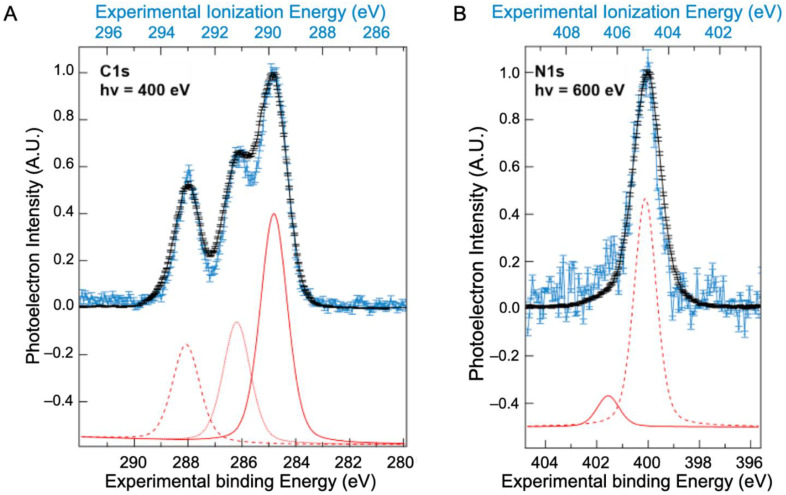
Comparison between the XPS spectra of BSA (**A**) C1s and (**B**) N1s levels measured on dry protein (black, bottom scale) and in solvated protein (blue, top scale). Contributions from atoms with different chemical shifts are shown in the fits to the solvated BSA below each spectrum (red). The energy difference between the binding energy and the ionization energy of the XPS signal is attributed to the work function of the aqueous solution (Φwater~4.9 ± 0.2 eV).

**Figure 3 ijms-23-08227-f003:**
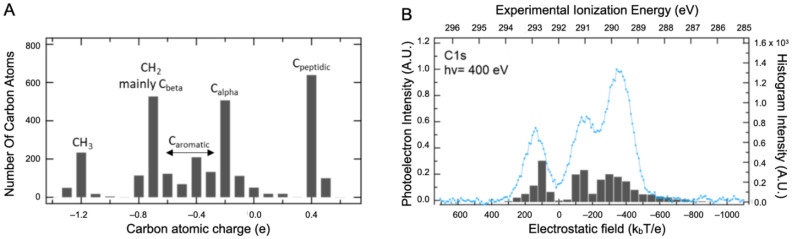
DFT calculation results. (**A**) Distribution of atomic charges calculated in BSA structure with a 3 Å water layer on its surface. (**B**) Comparison between the BSA experimental XPS signal recorded at 400 eV photon energy in the liquid state (blue line) and the local electric field felt by the different atoms (The gray bars).

**Figure 4 ijms-23-08227-f004:**
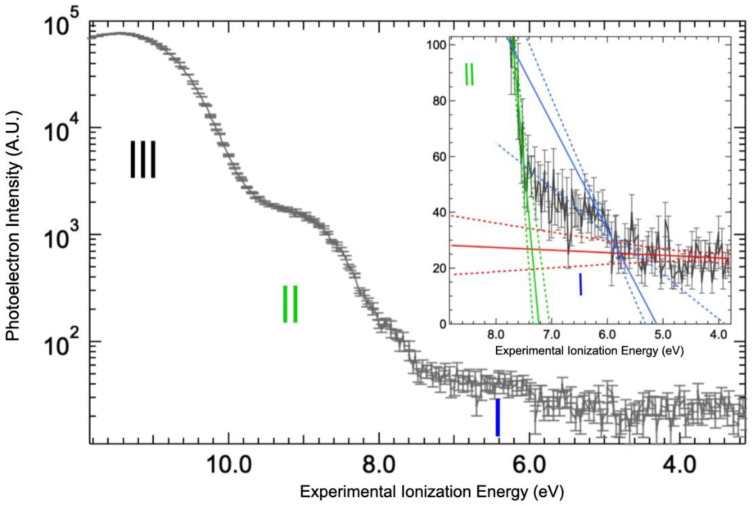
Valence bands for solvated BSA. The valence band threshold of the BSA aqueous solution was measured at 100 eV photon energy and represented by (black line), vertical bars corresponding to the statistical errors of the signals. Thresholds were determined by the intersection of linear fits (solid lines) with confidence intervals of 99% (dotted lines): background (red), band I (blue), and band II (green). The Y-axis of the main graph is in logarithm scale in order to clearly distinguish the three bands observed (identified with roman numbers).

**Figure 5 ijms-23-08227-f005:**
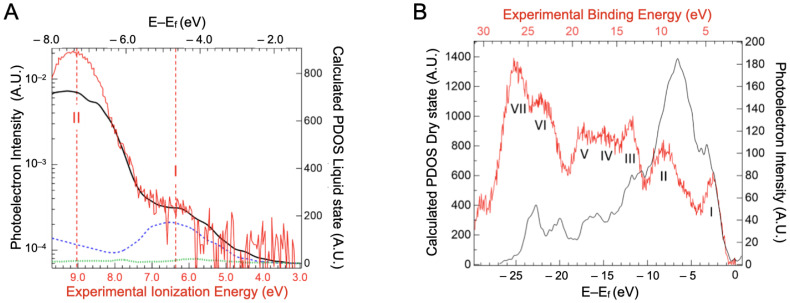
(**A**) Projected density of states of carbon (solid black line), nitrogen (blue dashed line), and sulfur (green dotted line) atoms derived from the calculated electronic structure of a BSA structure with 3 Å water over layer as a function of binding energy (top scale). Experimental data from valence spectra of the BSA aqueous solution are also presented (red line) with the contribution of the 1b1 ionization band of neat liquid water subtracted. (**B**) Projected density of state (PDOS) of the dry protein (black line) compared to the experimental valence band recorded on protein crystals (red line represented with statistical error). By comparison with density of state calculations performed on a dry protein, the band at −3 eV can be attributed to the unsaturated band (band I) and the one at −7 eV to the saturated band (band II). The calculated energies (E–E_f_) are referenced compared to the fermi level E_f_.

**Figure 6 ijms-23-08227-f006:**
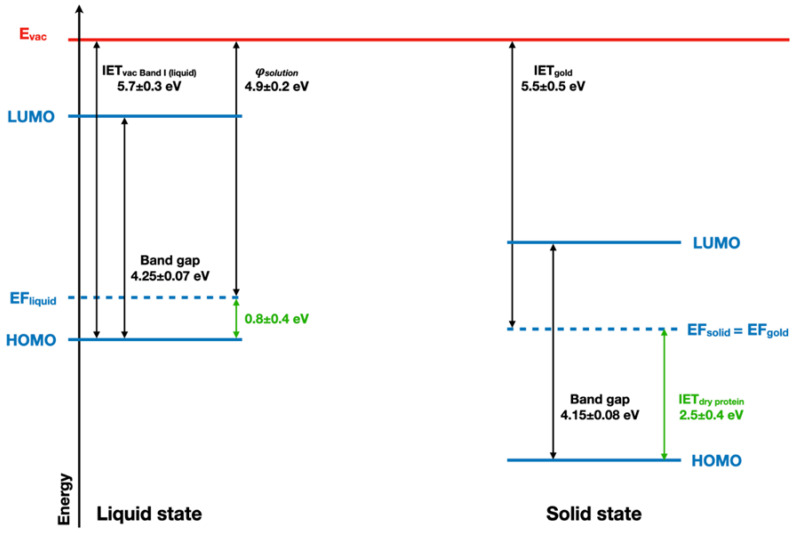
Schematic band structure of the solvated and dried BSA protein is considered as a semi-conductor.

**Table 1 ijms-23-08227-t001:** Comparison of spectral features measured in solid (dry) state and in the liquid state (solvated by water molecules).

	Solid State			Liquid State			Liquid State		C 1s Attribution and %At [10]	
	(This Study)			(This Study)			(NAP-XPS) [10]			
	Position (±0.35 eV)	%At	FWHM (eV)	Position (±0.13 eV)	%At	FWHM (eV)	Position (±1.7 eV)	FWHM (eV)		
C1s	284.9	47	1.18	289.8	52.5	0.93	285	1.47	C-C/C-H	46
	286.2	29	1.18	291.1	26	0.93	286.2	1.47	C-O/C-N	28
	288.0	24	1.18	293.0	21.5	0.93	287.4	1.47	N-C=O	24
							288.6	1.47	O-C=O	2

## Data Availability

The data presented in this study are available on request from the corresponding authors.

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
