# Peer review of "Electronic Structure and Solvation Effects from Core and Valence Photoelectron Spectroscopy of Serum Albumin"

_ijms, 2022, doi:10.3390/ijms23158227_

Round 1
Reviewer 1 Report
The manuscript by Renault et al. entitled “Electronic structure and solvation effects from core and valence photoelectron spectroscopy of serum albumin“ reports the use of XPS method to study electronic structure of bovine serum albumin. The manuscript is well written and I recommend its publication, although I suggest few editorial corrections:
1. Fig. 4 is unclear and in my opinion it should be bigger or divided into two figures.
2. It would be nice to supplement Conclusions with some examples of the application of such experiments, for example in which biological processes the knowledge about electronic structure is important.
3. There are few typos throughout the text, i.e. page 2, line 1 : "...the solutionis collected on a cold trap..."
Author Response
On behalf of my co-authors, I would like to thank you for your careful reading of our manuscript. We responded to all your constructive comments and suggestions, which allowed to improve the quality of our manuscript. In a general manner, we carefully re-read it and corrected different typos.Answers to the referee 1:
- Fig. 4 is unclear and in my opinion it should be bigger or divided into two figures.
We split the figure in two and updated the numbering of the following figures as well as the references to it.
- It would be nice to supplement Conclusions with some examples of the application of such experiments, for example in which biological processes the knowledge about electronic structure is important.
We added this sentence: “Moreover, it also establishes a methodology to determine, for a protein, the effect of the solvation on its electronic structure. This is of prime importance for biological processes, such as protein-ligand interactions (1), cofactor properties (2), and, more specifically, charge transfer in biosystems.”
- There are few typos throughout the text, i.e. page 2, line 1 : "...the solutionis collected on a cold trap..."
We added a space between “solution” and “is” and corrected many other typos (see word documents).
Reviewer 2 Report
The paper mainly presents two key contributions:
1. X-ray photoelectron spectroscopy study on Bovine Serum Albumin (BSA) on a liquid state
2. Study of charge transfer mechanism in a biological system
The characterization, and electronic structure study on solvation effects seemed to be logical making the technical and presentation aspects appear good enough. But still, there are some revisions which can be done to make the manuscript stronger. With suggested improvements along these lines, I think that the paper can be accepted in International Journal of Molecular Science. Comments and suggestion are listed below.
Comments
There are some unanswered questions which are listed below.
1. Page 4: The C1s spectra reported here is fitted with 3 peaks rather than typical 4 peaks. Why only 3 peaks used here? Has the author used 4 peaks for this peak fitting? Can the authors please explain in a better way?
2. Page 2: "studies, the solutionis collected....." - Please rephrase it correctly
3. Page 5: “Hydration leads to very limited changes in the charges”. – Can the authors explain this with relevant data?
4. Page 6: In this paper, electronic structure calculations were performed on a BSA structure with a 3 Angstrom water added layer. - Why 3 Angstrom water layer was used? How is the thickness of this water layer measured precisely? Does different thickness of water layer affect the calculations?
5. Page 7: Please redo the Figure 4 (B) and (C) with SI units in X and Y labels
6. Page 10: "All peak fittings were performed using a sum contribution Voigt-type function with 30% Lorentzian (SGL(30))". - Have the authors tried any other functions to fit the XPS spectra, e.g., GLP function (Product of Gaussian and Lorentzian)? Which peak shape best represents the XPS peaks of BSA?

Author Response
On behalf of my co-authors, I would like to thank you for your careful reading of our manuscript. We responded to all your constructive comments and suggestions, which allowed to improve the quality of our manuscript. In a general manner, we carefully re-read it and corrected different typos.
Answers to the referee 2:
- Page 4: The C1s spectra reported here is fitted with 3 peaks rather than typical 4 peaks. Why only 3 peaks used here? Has the author used 4 peaks for this peak fitting? Can the authors please explain in a better way?
We added this sentence: “The choice to use only three contributions is based on the higher constraints imposed by the high resolution XPS spectra obtained. Indeed, a fit with a fourth contribution will result, in our case, in areas that are not representative of the proportions of the most abundant types of carbon (see Table 1). Moreover, this choice is also supported by our calculations (see discussion below).”
- Page 2: "studies, the solutionis collected....." - Please rephrase it correctly
See the answer to the point 3 of referee 1.
- Page 5: “Hydration leads to very limited changes in the charges”. – Can the authors explain this with relevant data?
We added this sentence: “Indeed, the comparison of Hirschfeld charge calculations with water versus without added water displays a slope one that substantiates our affirmation (see the SI Appendix Fig. S9).” We also included the Figure S9 in the SI.
- Page 6: In this paper, electronic structure calculations were performed on a BSA structure with a 3 Angstrom water added layer. - Why 3 Angstrom water layer was used? How is the thickness of this water layer measured precisely? Does different thickness of water layer affect the calculations?
We added this sentence in the computation details sub-section: “The addition of a 3 Å water layer ensured conditions for the formation of a first continuous hydration shell (76). The layer was added using GROMACS, ensuring the precision in additions. We considered that this layer captured the main feature of hydrogen bond and water reorientation effects (77) and that the rest of the solution could be accounted for using solvent continuum approximation.
- Page 7: Please redo the Figure 4 (B) and (C) with SI units in X and Y labels
Figure 4 (B) and (C) now correspond to Figure 5 (A) and (B), according to the suggestion of referee 1 (point 1). The SI units have also been added.
- Page 10: "All peak fittings were performed using a sum contribution Voigt-type function with 30% Lorentzian (SGL(30))". - Have the authors tried any other functions to fit the XPS spectra, e.g., GLP function (Product of Gaussian and Lorentzian)? Which peak shape best represents the XPS peaks of BSA?
For this point we did not modify our manuscript, but rather comment directly to the referee 2. We chose the SGL(30) in order to directly compare our work with the previous ones on BSA. Even if the energy resolution of the XPS spectra of our study was never reached before, the structure remains broad in this type of nanomolecular edifice; also the two different numeric approximations of a Voigt profile give very similar results. Even a non-physical fit by a GL(30) does not affect the peak’s position and very slightly increases the FWHM by 0.03 eV.